# A pragmatic group sequential, placebo-controlled, randomised trial to determine the effectiveness of glyceryl trinitrate for retained placenta (GOT-IT): a study protocol

Fiona C Denison,[1] John Norrie,[2] Julia Lawton,[3] Jane E Norman,[1] Graham Scotland,[4] Gladys C McPherson,[2] Alison McDonald,[2] Mark Forrest,[2] Jemma Hudson,[2] Jane Brewin,[5] Mathilde Peace,[1] Cynthia Clarkson,[1] Sheonagh Brook-Smith,[6] Susan Morrow,[7] Nina Hallowell,[3,8] Laura Hodges,[1] Kathryn F Carruthers[1]

For numbered affiliations see end of article.

**Correspondence to**
Dr Fiona C Denison;
Fiona.Denison@ed.ac.uk

## ABSTRACT

**Introduction** A retained placenta is diagnosed when the placenta is not delivered following delivery of the baby. It is a major cause of postpartum haemorrhage and treated by the operative procedure of manual removal of placenta (MROP).

**Methods and analysis** The aim of this pragmatic, randomised, placebo-controlled, double-blind UK-wide trial, with an internal pilot and nested qualitative research to adjust strategies to refine delivery of the main trial, is to determine whether sublingual glyceryl trinitrate (GTN) is (or is not) clinically and cost-effective for (medical) management of retained placenta. The primary clinical outcome is need for MROP, defined as the placenta remaining undelivered 15 min poststudy treatment and/or being required within 15 min of treatment due to safety concerns. The primary safety outcome is measured blood loss between administration of treatment and transfer to the postnatal ward or other clinical area. The primary patient-sided outcome is satisfaction with treatment and a side effect profile. The primary economic outcome is net incremental costs (or cost savings) to the National Health Service of using GTN versus standard practice. Secondary outcomes are being measured over a range of clinical and economic domains. The primary outcomes will be analysed using linear models appropriate to the distribution of each outcome. Health service costs will be compared with multiple trial outcomes in a cost-consequence analysis of GTN versus standard practice.

**Ethics and dissemination** Ethical approval has been obtained from the North-East Newcastle & North Tyneside 2 Research Ethics Committee (13/NE/0339). Dissemination plans for the trial include the Health Technology Assessment Monograph, presentation at international scientific meetings and publication in high-impact, peer-reviewed journals.

**Trial registration number** ISRCTN88609453; Pre-results.

## Strengths and limitations of this study

► Pragmatic randomised placebo-controlled double-blind UK-wide trial to determine whether sublingual glyceryl trinitrate is (or is not) clinically and cost-effective for (medical) management of retained placenta.
► Inclusion of clinical, patient-sided, safety and economic outcomes with minimal exclusion criteria.
► Pragmatic approach to obtaining consent and good clinical practice (GCP)-training to facilitate recruitment in the emergency setting.
► Fully adaptive group sequential design.
► Inability to differentiate whether a placenta is adherent or trapped prior to administration of study drug.

## INTRODUCTION
### Background

A retained placenta (RP) is diagnosed when the placenta is not delivered within 30 or 60 min after delivery of the baby following active or physiological management of the third stage of labour, respectively[1] and is a major cause of postpartum haemorrhage.[2 3] The incidence of retained placenta is rising in the UK due to changes in maternal demographics, increased intrapartum interventions and operations involving exploration of the uterine cavity including surgical termination of pregnancy and dilation and curettage.[4] Retained placenta currently affects approximately 2% of vaginal deliveries[4] which equates to nearly 11 000 women per annum in the UK. Following failure of active or physiological management, the National Institute of Health and Care Excellence recommends

that retained placenta should be treated by the operative procedure of manual removal of placenta (MROP).[1] This procedure requires skilled personnel, an operative environment and, compared with spontaneous placental delivery, has its own attendant risks, including bleeding[3] and infection.[5] There is often a delay in undertaking the operation, which leads to further haemorrhage,[6] and the procedure itself is costly in terms of staff time. Furthermore, the infrastructure required for this operative intervention is not available in all delivery settings, and the invasive nature of this procedure has the potential to delay or interrupt mother–baby bonding in the immediate postpartum period. To reduce the number of women requiring MROP, there is therefore a need for new (and effective) medical treatments for retained placenta. The reduction in operative interventions would potentially have cost benefits for the National Health Service (NHS) and also for women in terms of increased satisfaction, less separation between mother and baby immediately after birth and reduced morbidity.

## Rationale for study

Observational studies along with results from small randomised controlled trials have suggested that nitric oxide donors including glyceryl trinitrate (GTN) given intravenously might be effective for the management of retained placenta with success rates ranging between 94% and 100%.[7–10] Although intravenous GTN appears to be efficacious for retained placenta in these small studies, this route of administration is not practical in all settings and intravenous administration causes unacceptable side-effects including symptomatic hypotension at higher doses.[3] More recently, sublingual administration has been trialled, with studies reporting experience with sublingual GTN tablet and spray. Using a sublingual GTN tablet, two small studies suggest a beneficial effect of sublingual GTN tablet; however, a third larger study does not (ISRCTN34755982; 37.3% vs 20.4%; GTN vs placebo; p>0.05; n=105 women).[11] The alternative preparation for sublingual administration is GTN spray.[12] Compared with the tablet preparation, sublingual GTN spray has several advantages including stability at room temperature, significant reduction in latency of onset with onset beginning at 30–45 s, peaking at 90–120 s and lasting up to 5 min,[13–15] and fewer objective and subjective side-effects.[16] GTN spray is an accepted obstetric management in other scenarios where rapid tocolysis is required such as uterine relaxation for release of a trapped head in breech delivery or at caesarean section.[12] Furthermore, anecdotal reports suggest that GTN spray may also have a use in management of retained placenta.[12]

To summarise, although a growing body of evidence supports GTN use for the treatment of retained placenta, much of this evidence is based on anecdotal case reports or clinical studies which are non-randomised, do not include a placebo arm and are underpowered. There is therefore a need to undertake a well-designed, randomised, placebo-controlled, double-blind pragmatic group sequential UK-wide (GOT-IT trial) to definitively determine whether sublingual GTN is (or is not) clinically and cost-effective for management of retained placenta.

## Study objectives

The aim of this placebo-controlled, double-blind, pragmatic group sequential UK-wide trial is to determine the effectiveness and cost-effectiveness of GTN for treating retained placenta and avoiding MROP.

The hypothesis being tested is that sublingual GTN spray is clinical and cost effective in treating retained placenta after vaginal delivery, reducing the need for MROP following failure of current management.

The primary objectives are to: (1) determine the clinical effectiveness of sublingual GTN in treating retained placenta and avoiding MROP in women with vaginal delivery following failure of current management (clinical domain), (2) determine the side-effect profile for GTN given to treat retained placenta (safety domain), (3) assess patient satisfaction with GTN given for retained placenta (patient-sided domain) and (4) assess the net costs (or cost savings) to the NHS of using GTN for the treatment of retained placenta compared with standard practice (economic domain).

The secondary objectives are to assess NHS costs in relation to the primary outcome and range of secondary outcomes which are expected to differ between the two arms of the trial, using a cost-consequence balance sheet approach.

The trial contained an internal pilot, which commenced in October 2014 and was completed in April 2015. The aim of this internal pilot, which recruited from eight sites across the UK, was to provide reassurance on all the trial processes, including recruitment, consent, randomisation, delivery of treatment and follow-up assessments to ensure that all were running smoothly. Recruitment to the substantive trial is ongoing and is due to finish in September 2017.

A nested qualitative study was undertaken during the internal pilot to adjust strategies to (1) maximise recruitment into the main trial, (2) refine the recruitment and consent pathway and (3) ensure staff are given appropriate training and support to help promote the successful delivery of the main trial. In-depth qualitative interviews were undertaken with staff (n=27) involved in trial recruitment, consenting and/or trial delivery and with trial recruits (n=22) within, when possible, 4 weeks of randomisation. The results of the qualitative research have been published and have informed the delivery of the main trial.[17 18]

## METHODS AND ANALYSIS
### Study design

This is a placebo-controlled, double-blind, pragmatic group sequential UK-wide trial to determine the effectiveness of GTN for treating retained placenta and avoiding MROP. Women are identified as being potentially eligible for the trial if, following vaginal birth, they have a retained placenta and are at risk of needing a MROP after failure

of current management of the third stage of labour. If eligible for the trial, women are treated with either the study drug or a placebo in a double-blind fashion. Following administration of the study medication, if the placenta still remains undelivered after 15 min, the treating clinician determines ongoing clinical management as per standard care.

The prespecified group sequential monitoring plan requires the Data Monitoring Committee to look at accumulating unblinded data on five occasions. The Data Monitoring Committee is using the results of these analyses to advise the Trial Steering Committee on adapting the trial design to either (1) stop prematurely for futility (no prospect of establishing a treatment effect of at least 10%) or (2) stop prematurely if proof beyond a reasonable doubt is established that there is a convincing treatment benefit of at least 10%. Assuming the trial recruits to its maximum size, we are aiming to recruit 1100 participants (maximum number randomised 1086, with the uplift to 1100 to account for a small allowance for withdrawal of consent or other rare loss to follow-up) over 42 months.

## Study population
### Setting
We are recruiting women from delivery wards in UK maternity hospitals. The delivery wards are of varying size and location ensuring the results of the trial will be generalisable to the UK. At present, recruitment is being undertaken in 28 sites across the UK.

### Selection of participants
As is standard practice, clinicians are undertaking the initial eligibility screening. Women are identified as being potentially eligible for the trial if, following vaginal birth, they have a retained placenta and are at risk of needing a MROP after failure of current management of the third stage of labour, defined as the placenta remaining undelivered after (1) 30 min of active management or (2) 60 min of physiological management plus a further 30 min of active management of the third stage of labour.

### Planned inclusion and exclusion criteria
All women will be considered for the trial who have RP after vaginal birth following failure of current management (defined as a third stage of labour lasting more than (1) 30 min after active management or (2) 60 min following physiological followed by 30 min of active management. In addition, they should also satisfy the following inclusion criteria. Should a woman have any of the exclusion criteria, they will not be invited to participate in the trial.

Women are eligible for inclusion if they are:
- ► Aged ≥16 years
- ► Vaginal delivery (including women with a previous caesarean section and an instrumental vaginal delivery in the delivery room)
- ► >14 weeks gestation

- ► Haemodynamically stable, defined as having a heart rate ≤119 beats per minute and a systolic blood pressure of >100 mm Hg

### Exclusion criteria
Women are excluded if they have one or more of the following:
- ► Unable to give informed consent
- ► Suspected placenta accreta/increta/percreta
- ► Multiple pregnancy
- ► Allergy or hypersensitivity to nitrates
- ► Consumption of alcohol in the last 24 hours
- ► Concomitant use of phosphodiesterase inhibitors
- ► Instrumental vaginal delivery in theatre
- ► Contraindications to nitrates including severe anaemia, constrictive pericarditis, extreme bradycardia, incipient glaucoma, glusose-6-phosphate dehydrogenase–deficiency, cerebral haemorrhage and brain trauma, aortic and/or mitral stenosis, angina caused by hypertrophic obstructive cardiomyopathy, circulatory collapse, cardiogenic shock and toxic pulmonary oedema

## Recruitment and study procedures
### Identifying participants
Recruitment was initiated in October 2014 and is currently ongoing in the delivery wards of 28 UK maternity hospitals. Recruitment is scheduled to conclude when 1100 subjects are recruited to the trial.

We initially followed the Royal College of Obstetrics and Gynaecology best practice recommendations for recruitment and obtaining consent for intrapartum research studies.[19] These recommendations were based on the work by Vernon et al,[20] who developed a consent pathway for women with retained placenta who were recruited to the RELEASE trial of umbilical vein oxytocin for RP.[21] We refined this pathway in light of our qualitative research[17 18] so that the consent and recruitment pathways were tailored to the specific requirements of this trial. In brief, where possible, outline information about the trial (including researcher contact details) is made available to women antenatally, particularly those identified as being at high risk of retained placenta, for example, maternal age >30 years, preterm birth and stillbirth.[22] More detailed information is available in antenatal clinics, on labour wards and the study website with information posters in the hospital reception areas and antenatal clinics. Following diagnosis of a retained placenta, a clinician or midwife who has received GOT-IT study training (which has sponsor-approved GCP-training embedded within it) invites women to participate in the trial, provides an explanation of the study and gives potential participants an abbreviated and/or full version of the patient information sheet.

### Informed consent
The patient information leaflets explain that the trial is investigating whether sublingual GTN spray is useful

for treating retained placenta. If a woman wishes to participate then written or verbal consent is obtained. If verbal consent is obtained, this is followed up by written consent as soon as possible in the postnatal period. Such an approach has been endorsed by the Royal College of Obstetricians and Gynaecologists in recent guidance about obtaining consent in perinatal research where consent is time critical in which it is stated that '[in acute circumstances]…provision of antenatal information to women with brief oral consent at the time of the complication is appropriate. Full written consent is then obtained at a later stage'.[23] Permission is sought to inform the woman's general practitioner that she is taking part in this trial. Due to the emergency nature of the condition (retained placenta), women are eligible to take part if they fulfil the inclusion criteria, understand what trial participation involves and give fully informed consent. A minimum time period is not feasible for trial consideration by potential participants. If a woman cannot give informed consent (eg, due to incapacity), she is not eligible for participation.

### Randomisation and allocation

The study drug is a metered dose sublingual spray containing 400 µg/dose of GTN and a matching placebo spray. The formulation consists of GTN, peppermint oil BP (88%), 1,1,1,2-tetrafluoroethane (HFA) and ethanol BP (96%). The placebo is identical to the active product minus the GTN. Both the study drug and the matching placebo are manufactured by Pharmasol Limited and labelled by Sharp Clinical Services (UK) Limited.

Study drugs are kept on the delivery suite and are provided to site pharmacies in prepacked randomised permuted blocks. After consent of each participant, the investigator or trained clinical practitioner randomises eligible and consenting women to one of the two study groups by taking the next box off the shelf. After priming the pump and demonstrating its use, the clinician gives the study drug to the women to self-administer. The study is double-masked, so neither the participant, investigator nor trained clinical practitioner know which treatment has been allocated. The boxes are numbered and labelled so that the identity of the drug can be determined later. The number on the box is logged in the study database. Breaking of the study masking is only performed where knowledge of the treatment is absolutely necessary for safe management of the patient.

A central emergency unblinding interactive voice response system is operated by the Centre for Healthcare Randomised Trials (ChaRT) in Aberdeen. Only senior clinicians can unblind trial participants. The name of the clinician requesting the unblinding, the reasons for it and notification of any unblinding is sent to the chief investigator via email. Reasons for unblinding are recorded in the participant's sources documents and in the Investigator Site Master File. Unless there is a clinical requirement, masking will not be broken until after all study data entry is complete, the validity of the data is

checked, all queries resolved and the patient populations agreed.

### Administration of study intervention

Women who have a retained placenta after vaginal birth or miscarriage following failure of management of the third stage of labour (failure of active management defined as the third stage lasting more than 30 min and failure of physiological management defined as the third stage lasting more than 60 min, plus a further 30 min of active management) are considered for eligibility for the trial.

Baseline observations (maternal blood pressure (mm Hg) and heart rate (beats per minute)) are taken prior to study drug administration to confirm that the woman fulfils the haemodynamic inclusion criteria (heart rate ≤119 bpm and systolic blood pressure >100 mm Hg) for trial entry.

Following confirmation of eligibility, women self-administer two puffs of either GTN (800 µg) or placebo spray as a once-only, sublingual intervention via a pump spray. Maternal temperature and a blood sample to measure haemoglobin are measured at baseline. Heart rate, systolic blood pressure and temperature (°C) are recorded at 5 and 15 min after administration of the study drug.

Sublingual GTN is maximally effective in causing uterine and cervical relaxation by 5 min postadministration. Thus, if spontaneous delivery of the placenta is going to occur, we expect this to happen around 5 min after drug administration. If spontaneous placental delivery has not occurred, a further attempt to deliver the placenta by controlled cord traction is made. If the placenta remains undelivered 15 min after administration of the treatment we think that it is highly unlikely that it will deliver spontaneously thereafter—we consider that any further delay risks haemorrhage. Thus, the decision that MROP is needed (primary clinical outcome) is made and the participant transferred to theatre for the definitive management of MROP as soon as possible. Additionally, if there are clinically significant side-effects (eg, haemorrhage, symptomatic maternal hypotension and/or tachycardia) before 15 min have elapsed after GTN/placebo treatment, the participant is transferred to theatre for immediate MROP.

### Follow-Up procedures

On the first postnatal day, a haemoglobin sample is collected. A questionnaire relating to the woman's perception of the study drug in relation to side-effects and patient satisfaction is also completed. A second questionnaire is sent by the Central Trial Office to participants at 6-week postdelivery. The questionnaire collects further information on patient-rated side-effects, patient-rated satisfaction and health economic resource use.

Women remain in the trial unless they are unable to continue for a clinical reason or choose to withdraw consent. All other changes in status, with the exception of

formal withdrawal of consent, mean the participant will be followed up for all study outcomes whenever possible.

## Safety

The laws that govern reporting for clinical trials of investigational medicinal products (IMP) regulate safety reporting for the GOT-IT trial. Adverse events (AEs) may occur during or after participation in the trial.

### Definitions

An AE is any untoward medical occurrence in a clinical trial participant which does not necessarily have a causal relationship with an IMP. Each AE is considered for severity, causality or expectedness and may be reclassified as a serious event.

An adverse reaction (AR) is any untoward and unintended response to the IMP which is related to any dose administered to that participant.

A serious adverse event (SAE) and serious adverse reaction (SAR) is any AE or AR that at any dose:

► results in death of the clinical trial participant;
► is life-threatening, defined as an event where the participant was at risk of death at the time of the event. It does not refer to an event which hypothetically might have caused death if it were more severe;
► requires inpatient hospitalisation or prolongation of existing hospitalisation;
► results in persistent or significant disability or incapacity;
► consists of a congenital anomaly or birth defect;
► results in any other significant medical event not meeting the criteria above.

A suspected unexpected serious adverse reaction (SUSAR) is any AR that is classified as serious and is suspected to be caused by the IMP, that it is not consistent with the information about the IMP in the summary of the medicinal product characteristics (SmPC).

### Identifying AEs and SAEs

All AEs and SAEs are recorded from the time a participant signs the consent form to take part in the study until the 6-week postnatal outcome assessment point.

Participants are asked about the occurrence of AEs/SAEs prior to discharge from the hospital and in the 6-week postnatal questionnaire. Open-ended and non-leading verbal questioning of the participant are used to enquire about AE/SAE occurrence at discharge. The 6-week postnatal questionnaire also asks participants if they have seen their general practitioner, been admitted to hospital or been prescribed any medication. If there is any doubt as to whether a clinical observation is an AE, the event is recorded. AEs and SAEs are also identified via information from support departments, for example, laboratories.

### Recording AEs and SAEs

When an AE/SAE occurs, it is the responsibility of the principal investigator to review all documentation (eg, hospital notes, laboratory and diagnostic reports) related to the event. The principal investigator records all relevant information on the SAE form (if the AE meets the criteria of SAE). Information to be collected includes type of event, onset date, Investigator assessment of severity and causality, date of resolution as well as treatment required, investigations needed and outcome.

The clinician will assess all reported SAEs. The events detailed in the list below are 'expected' and will not be reported to the cosponsors as an SAE but will be recorded in the case report form as a hospitalisation or outcome and presented to the Data Monitoring Committee, as part of the ongoing safety review.

In this trial, the following events are not considered SAEs: (1) pregnancy is not considered an AE or SAE, as it is part of the inclusion criteria; (2) hospitalisations for treatment planned prior to randomisation and hospitalisation for elective treatment of a pre-existing condition will not be considered as an SAE. This includes pregnancy; (3) fall in haemoglobin of more than 15% between recruitment and the first postnatal day; (4) MROP in theatre; (5) need for earlier than planned MROP on the basis of the clinical condition; (6) fall in systolic or diastolic blood pressure of more than 15 mm Hg and/or increase in pulse of more than 20 beats/min between baseline and 5 min and 15 min postadministration of active/placebo treatment; (7) need for blood transfusion between time of delivery and discharge from hospital; (8) need for general anaesthesia; (9) maternal fever (one or more temperature reading of more than 38°C within 72 hours of delivery or discharge from hospital if discharge occurs sooner); (10) sustained uterine relaxation after removal of placenta requiring uterotonics.

### Assessment of AEs and SAEs

Seriousness, causality, severity and expectedness of each AE are assessed by the principal investigator. AEs are assessed as though the participant is taking active IMP. Cases that are considered serious, possibly, probably or definitely related to IMP and unexpected (ie, SUSARs) are unblinded. This may be delegated to other suitably qualified physicians in the research team who are trained in recording and reporting AEs. The chief investigator may not downgrade an event that has been assessed by an investigator as an SAE or SUSAR, but can upgrade an AE to an SAE, SAR or SUSAR if appropriate.

#### *Assessment of seriousness*

The investigator makes an assessment of seriousness as defined in Definitions section.

#### *Assessment of causality*

The investigator makes an assessment of whether the AE/SAE is likely to be related to the IMP according to the definitions below.

► Unrelated: where an event is not considered to be related to the IMP.
► Possibly related: the nature of the event, the underlying medical condition, concomitant medication or

temporal relationship make it possible that the AE has a causal relationship to the study drug. The assessment of causality is made against the reference safety information found in the SmPC.

Where non-investigational medicinal products (NIMPs), for example, rescue/escape drugs are given: if the AE is considered to be related to an interaction between the IMP and the NIMP, or where the AE might be linked to either the IMP or the NIMP but cannot be clearly attributed to either one of these, the event will be considered as an AR. Alternative causes such as natural history of the underlying disease, other risk factors and the temporal relationship of the event to the treatment are considered and investigated; however, the blind will not be broken for the purpose of making this assessment.

### Assessment of expectedness

If an event is judged to be an AR, the evaluation of expectedness will be made based on knowledge of the reaction and the relevant product information documented in the SmPC. Events are classed as either expected (the AR is consistent with the toxicity of the IMP listed in the SmPC) or unexpected (the AR is not consistent with the toxicity in the SmPC).

### Assessment of severity

The investigator makes an assessment of severity for each AE/SAE and records this on the SAE form according to one of the following categories:

► Mild: an event that is easily tolerated by the participant, causing minimal discomfort and not interfering with every day activities.
► Moderate: an event that is sufficiently discomforting to interfere with normal everyday activities.
► Severe: an event that prevents normal everyday activities.

### Reporting of SAEs/SAR/SUSARs

Once the investigator becomes aware that an SAE has occurred in a study participant, the information is reported to the Academic and Clinical Central Office for Research and Development (ACCORD) Research Governance and Quality Assurance Office (Edinburgh University and NHS Lothian) immediately or within 24 hours. If the Investigator does not have all information regarding an SAE, they should not wait for this additional information before notifying ACCORD. The SAE report form can be updated when the additional information is received.

The SAE report provides an assessment of causality and expectedness at the time of the initial report to ACCORD. Where missing information has not been sent to ACCORD after an initial report, ACCORD contacts the investigator and requests the missing information until this is supplied.

All reports sent to ACCORD and any follow-up information are retained by the investigator in the Investigator Site File. ACCORD also informs the trial office who check that the SAE is added to the CHaRT database.

### Regulatory reporting requirements

The ACCORD Research Governance and Quality Assurance Office is responsible for pharmacovigilance reporting on behalf of the cosponsors (Edinburgh University and NHS Lothian). The ACCORD Research Governance and Quality Assurance Office has a legal responsibility to notify the regulatory competent authority and the Research Ethics Committee that approved the trial. Fatal or life-threatening SUSARs are reported no later than seven calendar days and all other SUSARs are reported no later than 15 calendar days after ACCORD is first aware of the reaction. ACCORD informs investigators at participating sites of all SUSARs and any other arising safety information. An Annual Safety Report/Development Safety Update Report is submitted, by ACCORD, to the regulatory authorities and Research Ethics Committees listing all SARs and SUSARs.

### Follow-up procedures

After initially recording an AE or recording and reporting an SAE, the investigator follows each participant until resolution or death of the participant. Follow-up information on an SAE is reported to the ACCORD office. AEs still present in participants at the 6-week postnatal outcome assessment point are monitored until resolution of the event or until no longer medically indicated.

## Outcome measures

The GOT-IT trial is assessing whether sublingual GTN spray is clinically and cost-effective in treating retained placenta after vaginal delivery, reducing the need for MROP following failure of current management.

### Primary outcome measures

The primary outcome is measured over four interrelated domains of clinical, safety, patient-sided and economic.

The primary clinical outcome is needed for MROP, defined as the placenta remaining undelivered 15 min poststudy treatment and/or being required within 15 min of treatment due to safety concerns.

The primary safety outcome is measured blood loss between administration treatment and transfer to the postnatal ward or other clinical area.

The primary patient-sided outcome is satisfaction with treatment, a side effect profile assessed by a questionnaire administered at discharge from hospital and a postal questionnaire sent to the participant 6-week posthospital discharge.

The primary economic outcome is net incremental costs (or cost savings) to the NHS of using GTN versus standard practice. Costs include GTN (dose and time to administer drug), monitoring of the woman and delivering the placenta if effective, MROP if required and further health service resource use to 6 weeks postdischarge.

### Secondary outcomes

The secondary endpoints are measured over two interrelated domains of clinical and economic outcomes. There are no secondary patient-sided or safety outcomes.

The secondary clinical endpoints include: (1) fall in haemoglobin of more than 15% between recruitment and the first postnatal day; (2) time from randomisation to delivery of placenta; (3) MROP in theatre; (4) need for earlier than planned MROP on the basis of the clinical condition, (5) fall in systolic or diastolic blood pressure of more than 15 mm Hg and/or increase in pulse of more than 20 beats/minute between baseline and 5 min and 15 min postadministration of active/placebo treatment; (6) need for blood transfusion between time of delivery and discharge from hospital; (7) need for general anaesthesia; (8) maternal fever (one or more temperature reading of more than 38°C within 72 hours of delivery or discharge from hospital if discharge occurs sooner); (9) sustained uterine relaxation after removal placenta requiring uterotonics; (10) readmission to hospital for any reason within 6-week postpartum.

Estimated health service costs will be compared with multiple trial outcomes in a cost-consequence analysis of GTN versus standard practice. The costs and consequences of the alternative approaches to treatment will be summarised using a balance sheet, highlighting the costs and outcomes favouring each treatment option.

## Data collection and processing
### Collection of outcome data
The primary clinical outcome (need for MROP, defined as the placenta remaining undelivered 15 min post-treatment and/or being required within 15 min of treatment due to safety concerns) is collected 15 min postadministration of the study drug. The primary safety outcome (measured blood loss between administration treatment and transfer to the postnatal ward/other clinical area) is recorded immediately prior to transfer from theatre to the postnatal ward or other clinical area (eg, labour ward high dependency). The primary patient satisfaction outcome is collected at 6 weeks by a self-completed satisfaction questionnaire and the primary economic outcome will be computed from costs incurred at initial management of retained placenta and management of further complications and self-reported health service usage and side-effects recorded from the postnatal patient questionnaires.

Outcome data are collected on an electronic case report form and paper-based questionnaires by research midwives and nurses who have been specifically trained in data collection for the trial. The electronic case record form comprises the following sections: baseline data, clinical observations and placenta delivery, first post natal day, discharge, 6-week postnatal check, safety events, discharge and follow-up questionnaires. Health service use in the 6 weeks following discharge is collected via a small number of resource use questions included in the postnatal questionnaires. Resource use will be evaluated using routine sources of nationally relevant unit costs. Women are permitted to receive all standard care medical treatments for postpartum complications including any given for medical management of retained placenta.

Concomitant medications are recorded from 24 hours prior to signing of the consent forms until 24 hours after administration of the study drug. Analgesic or antibiotic medications are recorded from 24 hours prior to signing of informed consent until 6 weeks after administration of study drug.

## Sample size, proposed recruitment rate and milestones
### Sample size
The trial uses a group sequential design as opposed to a fixed sample design to allow for the uncertainty of two critical factors. First, there is considerable uncertainty about how many women who are eligible for the trial will actually go on and require surgery due to (1) lack of knowledge of the frequency of spontaneous delivery of the placenta beyond the time frame that GTN is known to have a pharmacological effect and (2) the effect of variations in local clinical practice in relation to organising MROPs. Second, the magnitude of the benefit from GTN spray is unknown. To properly reflect these uncertainties, we are using a group sequential design. This design maximises our chances of efficiently detecting and estimating the true benefit of treatment in the quickest time with the right number of participants and gives us the opportunity to abandon the trial if it turns out that no worthwhile treatment effect exists (via futility analyses).

In discussion with clinicians and women, a minimally clinically important difference of an absolute reduction of 10% in the need for MROP would need to be observed in order to accommodate this intervention (sublingual GTN) in clinical practice. Given that the maximum variability in a binary outcome (yes or no) occurs at a 50% rate in the placebo spray arm, a fixed sample approach would need 1038 women (519 in each group) to demonstrate a 10% change from 50% on placebo to 40% on GTN spray. As the outcome (surgery occurred—yes or no) takes place 15 min after the intervention has been administered, there will be minimal loss to follow-up and, therefore, no requirement for a trial much larger than this. Using a group sequential design incorporating a Lan-DeMets alpha spending function with O'Brien-Fleming boundaries, our trial has a maximum sample size of 1086 (with a small inflation to 1100 to allow for withdrawal of consent of other rare loss to follow-up). This allows for five planned interim evaluations by the Data Monitoring Committee of accruing data at 218, 436, 652 and 870 participants with a final look (if the trial proceeds to full size) at 1086 women.

## Analyses plan
### Statistical analysis
The statistical analyses will be according to the intention-to-treat principle and will be governed by a statistical analysis plan which will govern all statistical aspects of the study, including the statistical methods for the analysis of all the outcomes, approaches to missing data and sensitivity type analyses to investigate the robustness of any findings reported. The statistical analysis plan will be

finalised (including the decision to undertake prespecified subgroup analyses or not) prior to the data being locked for final analysis. The interim analyses will be specified within the Data Monitoring Committee Charter and the result of the interim analyses will be in strict confidence and unknown to the trial research group apart from the study statistician. The primary outcomes, clinical (MROP), safety (blood loss), patient satisfaction and economic (NHS costs), will be analysed using generalised linear models appropriate to the distribution of each outcome. These models will adjust for relevant baseline factors, and sensitivity analyses will assess the robustness of the findings to any missing data (which is expected to be very low as the primary outcome is measured within minutes of receiving the intervention). The interim analysis will be performed in East[24] and all remaining analysis will be done in Stata.[25] Significance will be defined as $p<0.05$. In the event prespecified formal subgroup analyses of the primary outcome are undertaken, we will formally assess subgroup(s) by fitting a subgroup * randomised group interaction term and these interactions will be assessed against a stricter level of significance of $p<0.01$ to guard against chance findings.

### Economic evaluation

While the cost of GTN is low, there will still be costs associated with its administration and the monitoring and management of women thereafter. Should the intervention prove effective, it will be important in the context of scarce maternity resources to explicitly quantify the net costs (or cost savings) associated with its use. We have therefore made provision to carry out a simple cost analysis using clinical and resource use data, which is being collected for individual participants recruited to the trial. This analysis will explicitly quantify the difference in mean costs between the active intervention and placebo arm. Research costs associated with placebo delivery will be factored out of the analysis to estimate the incremental cost (or cost savings) of the active intervention versus standard practice.

Resource use associated with the alternative management strategies will be estimated from the time of randomisation through to 6 weeks postpartum. This will include: (1) staff time for administering the drug to patients; (2) resource use associated with any complications arising following administration of the study drug (eg, blood pressure and/or heart rate monitoring); (3) subsequent costs associated with delivery of the placenta (either spontaneously or operatively) and (4) subsequent health service contact relating to retained products of conception up to 6 weeks postdischarge. Resource use collected for individual participants will be valued using accepted national unit costs for drugs, staff time, MROP and subsequent healthcare contacts.[26–28] The total cost of care for each participant will be calculated for each individual.

The mean costs will be summarised by treatment allocation group, and the incremental cost (or cost saving) associated with the use of GTN will be estimated using an appropriately specified general linear model. The cost data will be presented alongside the primary and secondary outcome data in a cost-consequence balance sheet, indicating which strategy each outcome favours.[29]

## Organisation: trial management, oversight and governance arrangements

### Trial management

The trial is being jointly run by CHaRT based at the University of Aberdeen, the chief investigator and the ACCORD office at the University of Edinburgh. The trial management and governance are as recommended by The Research Governance Framework and The Medicines for Human Use (Clinical Trials) Regulations 2004. Briefly, we have set up a Trial Steering Committee and a Data Monitoring Committee led by individuals independent of the project and the institutions involved. The membership of the Trial Steering Committee includes independent members including a lay representative. A separate and independent Data Monitoring Committee has also been convened. The Data Monitoring Committee met during trial set-up to agree terms of reference and is meeting a further five times during recruitment to monitor accumulating data and oversee safety issues. The Data Monitoring Committee will meet one final time after unblinding to assess overall data validity and analyses. The Trial Steering Committee is meeting approximately seven times with the meeting spacing being matched to that of the Data Monitoring Committee. Finally, we have regular Project Management Group meetings and Research Midwives Meetings to ensure consistency of trial processes across sites and share best practice for recruitment, consent and data collection.

### Research governance and compliance
*Research governance*

ACCORD is providing oversight that the appropriate research and competent authority approvals are in place, pharmacovigilance services and is ensuring the trial is conducted in accordance with the principles of the International Conference on Harmonisation Tripartite Guideline for Good Clinical Practice.

*Data protection*

The trial complies with the Data Protection Act 1998 and regular monitoring and checks are in place to ensure compliance. Data are being stored in accordance with the Act and will be archived to a secure data storage facility. The consent form states that other researchers may wish to access (anonymised) data in the future. The senior information technology manager (in collaboration with the trial statistician and trial manager) will manage access rights to the data set. Prospective new users must demonstrate compliance with data protection, legal and ethical guidelines before any data are released. We anticipate that anonymised trial data will be shared with other researchers to enable international prospective meta-analyses.

*Sponsorship*

The University of Edinburgh and NHS Lothian are the cosponsors for the trial. Cosponsorship responsibilities

are detailed in an agreement. Responsibilities subcontracted to CHaRT are also detailed in an agreement which detail the responsibilities surrounding centralised trial administration, database support and economic and statistical analyses.

### Quality assurance

An independent risk assessment has been performed to determine the level of monitoring required and if an audit should be performed before/during/after the study and if so, at what locations and at what frequency.

### Data handling, record keeping and archiving

The local principal investigator or research midwives enter locally collected data from the study centres into the web-based data collection application. The 6-week follow-up questionnaires are sent from and returned to the Study Office in Edinburgh and are input by the staff in the Study Office in Edinburgh. CHaRT is working with the Study Office in Edinburgh and with local Research Midwives to ensure that the data are as complete and accurate as possible.

## ETHICS, REGULATORY APPROVALS AND DISSEMINATION
### Ethics and regulatory approvals

The trial is registered on the International Standard Randomised Controlled Trial Number (ISRCTN) as ISCRTN88609453. Ethical approval has been obtained from the North East- Newcastle & North Tyneside 2 Research Ethics Committee (13/NE/0339). Approval has also been obtained from the Medicines and Healthcare Products Regulatory Authority (MHRA) (2013-003819-42) and the Health Research Authority as well as approvals from the local Trust Research and Development Offices. The study has been adopted to the National Institute for Health Research Clinical Research Network (NIHR CRN) Portfolio (http://www.nihr.ac.uk/research-and-impact/nihr-clinical-research-network-portfolio/). The necessary trial insurance is provided by the University of Edinburgh.

### Dissemination

The dissemination plans include the Health Technology Assessment monograph, presentation at international scientific meetings and publication of the results of the main trial and each of the secondary endpoints in high impact peer reviewed journals. If all grant-holders and researcher staff fulfil authorship rules, group authorship will be used under the collective title of 'the GOT-IT Trial Group'. The chief investigator, trial manager and possibly other members of the trial group will take responsibility for drafting the paper and this will be recognised by line 'the chief investigator (as primary author), followed by the other authors and the GOT-IT Trial Group'. Authorship of the qualitative substudy publications[17 18] has been attributed to chief investigator and the named individual(s). To maintain interest in the study, we are also publishing GOT-IT newsletters at intervals to trial staff and collaborators.

**Author affiliations**
[1]Tommy's Centre for Maternal and Fetal Health Research, Medical Research Council Centre for Reproductive Health, Queen's Medical Research Institute, University of Edinburgh, Edinburgh, UK
[2]The Centre for Healthcare Randomised Trials (CHaRT), Health Services Research Unit, University of Aberdeen, Aberdeen, UK
[3]Centre for Population Health Sciences, University of Edinburgh, Edinburgh, UK
[4]Health Economics Research Unit, Institute of Applied Health Sciences, University of Aberdeen, Aberdeen, UK
[5]Tommy's, Nicholas House, London, UK
[6]Simpson's Centre for Reproductive Health, Royal Infirmary of Edinburgh, Edinburgh, UK
[7]Usher Institute of Population Health Sciences and Informatics, University of Edinburgh, Edinburgh, UK
[8]Ethox Centre, Nuffield Department of Population Health, University of Oxford, Oxford, UK

**Contributors** FD conceived and designed the study. FD, JN, JL, GS, GCM, AM, JB, MP, CC, SBS, NH, LH and JEN drafted the original grant proposal and trial protocol. JN and JH provide methodological and statistical expertise. JL and NH provide expertise in the qualitative studies. FD, JEN and SBS provide expertise in the clinical outcomes. LH prepared the original trial protocol and ethics application and KFC and SM have drafted protocol revisions. LH, SM and KFC have all been trial managers for the trial. MF and AM provide database and trial support from the trials unit. JB and CC provide lay input. FD and KFC drafted the manuscript. KFC has responsibilities for day-to-day running of the trial including participant recruitment, data collection and liaising with other sites. All authors critically reviewed and approved the final version of the manuscript.

**Funding** The GOT-IT Trial (trial registration number: ISCRTN 88609453) is funded by the National Institute for Health Research Health Technology Assessment (HTA) programme (Project number 12/29/01). The views and opinions expressed therein are those of the authors and do not necessarily reflect those of the HTA, the National Institute for Health Research, the NHS or the Department of Health. This work was undertaken in the MRC Centre for Reproductive Health which is funded by MRC Centre grant (MRC G1002033).

**Competing interests** None declared.

**Patient consent** Obtained.

**Ethics approval** Ethical approval has been obtained from the North East-Newcastle & North Tyneside 2 Research Ethics Committee (13/NE/0339).

**Provenance and peer review** Not commissioned; externally peer reviewed.

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
