## [Reviewer comments · BMJ Open]

ARTICLE DETAILS

TITLE (PROVISIONAL)	A pragmatic group sequential placebo controlled randomised trial to determine the effectiveness of G lyceryl trinitrate for r etained placenta (GOT-IT): a study protocol
AUTHORS	Denison, Fiona; Norrie, John; Lawton, Julia; Norman, Jane; Scotland, Graham; McPherson, Gladys; McDonald, Alison; Forrest, Mark; Hudson, Jemma; Brewin, Jane; Peace, Mathilde; Clarkson, Cynthia; Brook-Smith, Sheonagh; Morrow, Susan; Hallowell, Nina; Hodges, Laura; Carruthers, Kathryn

VERSION 1 - REVIEW

REVIEWER	Blake Rodgers St Louis University Family Medicine Residency United States
REVIEW RETURNED	17-Apr-2017

GENERAL COMMENTS	There is minimal discussion of the limitations of the study. Some exclusion criteria are vague (page 11, line 9-12) such as severe anaemia and extreme bradycardia. The study would be stronger to specify what levels constitute a contraindication to nitrates and therefore will not be included. Otherwise it is up to individual investigator judgment. Page 11, line 45 - consider specifying your high risk criteria for identifying patients at high risk of retained placenta. As stated in the protocol this is left up to the judgment of the individual site investigators. This could lead bias (albeit likely very small) as some women are given information prior to delivery and some are not based on judgment and not consistent across research sites. Page 13, line 38 - consider recording heart rate and blood pressure longer than 15 minutes since GTN effects can last 25 minutes.
---

REVIEWER	Prof.Hany Abdel-Aleem Department of Obstetrics and Gynecology , Faculty of Medicine , Assiut University , Assiut, Egypt "None declared " . But lam an author of a Cochrane systematic review on the same topic
REVIEW RETURNED	20-Apr-2017

GENERAL COMMENTS	Abstract : Page 2 : Methods and analysis : you are mentioning the outcomes .Here you have to mention briefly the design methods and analysis of your data !.
---

1.2 Rationale for study

-Page 5,6: [Observational studies have suggested that nitric oxide donors including glyceryl trinitrate (GTN) given intravenously, might be effective for the management of retained placenta with success rates ranging between 94% to 100% [7-10]. Reference 7 is related to a Cochrane systematic review including three RCTs and not Observational studies . Either to move the reference from here or to add RCTs and observational studies .

Page 6, Row 34: [To summarise, although a growing body of evidence supports GTN use for the treatment of retained placenta, much of this evidence is based on anecdotal case-reports or clinical studies which are nonrandomised, do not included a placebo arm and are underpowered]

-I don't agree with this summary because the available data based on systematic review (7) showed that the use of NTG alone did not reduce the need for manual removal of placenta. And there is a need for well-designed RCT , placebo controlled to test the role of nitroglycerine in the management of retained placenta .

1.3 Study objectives : Ok

2.1 Study design

Page 9 , row 8 : [The aim of this placebo controlled, double blind, pragmatic group sequential UK-wide trial is to determine the effectiveness of GTN for treating retained placenta and avoiding manual removal of placenta] .

- Don't repeat the aim , just this is a placebo controlled, double blind, pragmatic group sequential UK-wide trial.

Page 8, row 25-40: Here you mention why pragmatic group sequential is the best design for your trial ,. What are the shortcomings if any . Why you are stressing on the stopping rules only ! .

2.2 Study Population

2.2.1 Setting.

Page 8, row 54: It is good that the sites are diverse to be representative to UK , but what is the level of these settings !, elaborate on this to see if these are comparable or could be a

source of heterogeneity .

2.2.3 Planned inclusion and exclusion criteria

Page 9 :Inclusion criteria:

- You did not refer to third stage management [active or expectant]. I think you will include both, clarify this in the inclusion criteria

- You did not refer to whether the placenta is detached or not! .

-[>14 weeks gestation], this means that you will include cases of miscarriage!. This is not vaginal birth, clarify this !.

Exclusion Criteria: Clarify how you will ascertain the absence /presence of these excluding factors !

-Bleeding should be an exclusion criteria because this is an indication to intervene ! .

2.3.1 Identifying participants

Page 10 . All the paragraph describes the consent rather than recruitment. You refer to risk factors for RP without mentioning these risk factors ! .

2.3.3 Randomisation and allocation

Page 11. All the paragraph about allocation and not randomization. What is method of randomization , do you make stratification by center ! .

2.3.4 Follow up procedures

Page 12 : you have to describe the intervention separate from follow up .You will include cases of RP after vaginal birth and miscarriage . You will include cases managed actively or expectantly in the third stage . You have to describe what you will do in each clinical scenario from the time of diagnosis, then the intervention and the follow up of cases .

-I wonder if you will examine the cases vaginally to see if the placenta is detached or not ! . You will do ultrasound examination or not ! .

-The intervention will be NTG spray only or there will be oxytocin drip ! . Specify the Intervention, manufacturing company etc.

-I wonder if there will be any uterine massage after delivery of the baby !

-The time limits mentioned will be applied to cases of RP following miscarriage! .

-I wonder if you will measure blood loss after delivery of the baby till delivery of the placenta or not! .

-I wonder if starting breast feeding immediately (before delivery of the placenta) will be allowable!

-You will include two types of questionnaire (one related to safety and the other to satisfaction), either mention details related to content, or alternatively add as annex.

2.4 Safety

Page 13: Definitions and Details of reporting safety issues are mentioned in the manual of operations of the trial or SOPs . In the protocol , you put that you are following GCP procedures with a reference .

2.5.1 Primary outcome measures

-Page 19:[The primary safety outcome is measured blood loss between administration treatment and transfer to the postnatal ward or other clinical area.]. Mention how you will measure the blood loss in the follow up procedures.

-Page 20:[The primary economic outcome]. Mention more details on how to calculate this outcome

2.5.2 Secondary outcomes:

- Manual removal of the placenta is a primary outcome!

-Use of general anesthesia is linked to manual removal of the placenta or you will use it for other condition!

-[sustained uterine relaxation after removal placenta requiring uterotonics]. You should mention what is the routine practice after delivery of the placenta , to differentiate it from additional uterotonics!.

-Fall in hemoglobin and fall in blood pressure are safety issues!. As you know NTG is a nitric oxide donor with an effective smooth muscle relaxant with a potent and short-lived tocolytic effect .

-Maternal pyrexia per se is not an outcome, unless it is related to infection as a consequence of MOP. It is better to put need for treatment with antibiotics.

-Subsequent surgical evacuation of retained products of conception. NTG could reduce the need for MOP but increase in the need for surgical evacuation of retained products .

2.6 Data Collection and Processing

2.6.1 Measuring outcomes , the title is measuring outcomes, but the content don't reflect this !.

Page 21:[The primary safety outcome (measured blood loss between administration treatment and transfer to the postnatal ward/other clinical area) is recorded immediately prior to transfer from theatre to the postnatal ward or other clinical area (e.g. labour ward high dependency). **How you will measure the blood loss!**

-[The primary patient satisfaction outcome is collected at six weeks by a self-completed satisfaction questionnaire] **put the questionnaire as annex**

-[Estimated health service costs will be compared with multiple trial outcomes in a cost-consequence analysis of GTN versus standard practice. The costs and consequences of the alternative approaches to

treatment will be summarised using a balance sheet, highlighting the costs and outcomes favouring each treatment option].**Put more details on this cost-sequence analysis or put reference]**.

Page 21, row 24[Outcome data is collected on electronic case report forms and paper-based questionnaires]. **Your data collecting tools should be added as annex or you mention the main information collected as text .**

2.7.1 Sample size

Page 21: Though the magnitude of effect of NTG spray per se is not known , but the effect of NTG is reported in at least three RCTs .This could be a guide to calculate the sample size .

2.8.1 Statistical Analysis

**What is the software, you will use for data entry and analysis?
What statistical methods you will use for different comparisons !
What is the level of significance?**

Page 23, row 11[Subgroup analyses of the primary outcome will be explored.]Specify **what subgroup analyses you will explore!**

VERSION 1 – AUTHOR RESPONSE

Reviewer: 1

C1: There is minimal discussion of the limitations of the study.

R1: We have followed the guidelines for protocol paper submission to BMJ open. These guidelines do not require inclusion of a discussion, which is where study strengths and limitations would be discussed. We believe we can only fully address the limitations at the end of the trial. If we do this at the stage of protocol submission or midway through the trial (as we are currently), we will be ignorant of many of the limitations and risk boosting the strengths of the trial. We will discuss the strengths and weaknesses of the trial in full when we submit the completed trial for publication when we will be in a better position to fully evaluate them and discuss in a balanced way.

C2: Some exclusion criteria are vague (page10, line207-208) such as severe anaemia and extreme bradycardia. The study would be stronger to specify what levels constitute a contraindication to nitrates and therefore will not be included. Otherwise it is up to individual investigator judgment.

R2: Our trial is pragmatic with trial processes embedded as far as possible within clinical practice. When a woman develops the obstetric emergency of retained placenta, they are seen and assessed by clinicians who use their clinical judgment about ongoing care. It is therefore appropriate that the decision to include or exclude a woman from trial entry is taken by attending clinicians because this is what happens in clinical practice. The exclusion criteria are based on the exclusion criteria from the Summary of the Product Characteristics for GTN. Severe anaemia and extreme bradycardia are not defined in the SmPC. Retained placenta is an obstetric emergency which needs urgent management. Specifying an exact haemoglobin value for severe anaemia would require taking a maternal blood sample at diagnosis of retained placenta and waiting for the result. Delaying treatment would increase the risk of bleeding and which would be unsafe and unethical. In the case of extreme bradycardia, this would likely result in cardiovascular instability which would exclude the woman from taking part.

C3: Page 13, line 287 - consider recording heart rate and blood pressure longer than 15 minutes since GTN effects can last 25 minutes.

R3: Sublingual GTN has a rapid onset of activity (within 1 minute) and has a plasma half-life of 1-3 minutes. Although the bioavailability, pharmacokinetics and pharmacodynamics of sublingual GTN are variable, clinically significant haemodynamic effects are maximal within the first 15 minutes. We therefore chose to measure these parameters at zero, five and 15 minutes which accord with monitoring, associated with the clinical care provided to women with this obstetric emergency.

Reviewer: 2

C1: Abstract : Page 2 : Methods and analysis : you are mentioning the outcomes. Here you have to mention briefly the design methods and analysis of your data

R1: As requested, we have amended the abstract (lines 31 – 34 and 39-40).

C2: 1.2 Rationale for study Page 5,6: [Observational studies have suggested that nitric oxide donors including glyceryl trinitrate (GTN) given intravenously, might be effective for the management of retained placenta with success rates ranging between 94% to 100% [7-10]. Reference 7 is related to a Cochrane systematic review including three RCTs and not Observational studies. Either to move the reference from here or to add RCTs and observational studies.

R2: We thank the reviewer for their helpful comment indicating that the Cochrane systematic review includes three RCTs (although we note that two were excluded from the review). We therefore believe that the reviewer is not referring to reference 7 in the manuscript, but instead to the updated Cochrane Review published in 2015 which does report results from 3 RCTs. We have amended the text to reflect the inclusion of these RCTs as well as observational studies and have updated reference 7 so that it now refers to the updated Cochrane Review published in 2015 as opposed to the earlier review, which was cited in the original protocol paper submission.

C3: Page 6, Row 113: [To summarise, although a growing body of evidence supports GTN use for the treatment of retained placenta, much of this evidence is based on anecdotal case-reports or clinical studies which are nonrandomised, do not included a placebo arm and are underpowered] - I don't agree with this summary because the available data based on systematic review (7) showed that the use of NTG alone did not reduce the need for manual removal of placenta. And there is a need for well-designed RCT, placebo controlled to test the role of nitroglycerine in the management of retained placenta.

R3: While we respectfully note that the reviewer does not agree with our summary describing the lack of evidence based support for use of GTN for the treatment of retained placenta, we do not feel that the data in the systematic review (Ref. 7) provides sufficiently robust evidence to allow this conclusion to be reached. As we mentioned in our response to C2, we believe that the reviewer's comments may refer to the updated 2015 review which reports on outcomes for 3 randomised controlled trials which have a combined total of 175 women with retained placenta. Of note, although the primary outcome was reported for all 3 trials (n=175), no significant differences were noted between the GTN and placebo groups in reducing the need for manual removal of placenta. However one of these trials (n=111) was reported as being at high risk of bias due to incomplete outcome data and selective reporting. Furthermore, the primary outcome was not reported for the two remaining RCTs (n=64 women), which were described as being at low risk of bias. Therefore, we believe that inclusion of results from a trial which has been reported of being at high risk of bias, as well as the small size of the trials which were reported, do not justify the reviewer's conclusion that that there is no benefit derived from use of GTN in the treatment group, and the NIHR/HTA agreed that this was a topic

needing a definitive high quality randomized trial to inform the appropriate treatment for women in the UK NHS when they commissioned this GOT-IT trial with a £1.7m budget. We have revised the manuscript in light of the reviewers comment and have emphasized our agreement with the reviewer that there is a need for a well-designed, large scale RCT to address this question.

C4: 1.3 Study objectives : Ok

R4: Thank you.

C5: Page 8 , row 152 : *[The aim of this placebo controlled, double blind, pragmatic group sequential UK-wide trial is to determine the effectiveness of GTN for treating retained placenta and avoiding manual removal of placenta]. Don't repeat the aim, just this is a placebo controlled, double blind, pragmatic group sequential UK-wide trial.*

R5: We have addressed this point and removed "The aim of."

C6: Page 8, rows 160-165: *Here you mention why pragmatic group sequential is the best design for your trial. What are the shortcomings if any. Why you are stressing on the stopping rules only.*

R6: We felt the group sequential design was the most appropriate since as discussed above there was considerable uncertainty around both the primary event rate in the control group (the untreated rate) and also the likely magnitude of the treatment benefit if one existed given the sparsity and poor quality of the literature. The group sequential design with 5 scheduled formal interim analyses provided the flexibility to declare that the question was answered decisively before recruiting to the larger fixed sample size driven by the assumed minimal important clinical difference. The group sequential approach requires only a modest uplift (<10%) in the potential maximum sample size (compared with the fixed sample), with the only disadvantages of a group sequential approach being the complexity – both in organizing the meetings of the iDMC and also (marginally) in analyzing and reporting and interpreting the findings.

C7: Page 8, row 171-172: *It is good that the sites are diverse to be representative to UK, but what is the level of these settings, elaborate on this to see if these are comparable or could be a source of heterogeneity.*

R7: Our trial is a pragmatic Phase 4 trial which is evaluating whether sublingual GTN is clinically and cost-effective for treating retained placenta in real as opposed to ideal clinical circumstances. Thus, recruiting women from delivery units of different sizes is a strength and not a weakness of this study,

C8: Page 9 : *Inclusion criteria: You did not refer to third stage management [active or expectant] . I*

think you will include both, clarify this in the inclusion criteria. You did not refer to whether the placenta is detached or not. [>14 weeks gestation], this means that you will include cases of miscarriage. This is not vaginal birth, clarify this.

R8: We do include both active and physiological third stages management in the trial and have now clarified this in the text. As this is a pragmatic trial we do not wish to delay the intervention. We do not perform an ultrasound scan to verify if the placenta is detached or non-detached because there is a paucity of evidence that ultrasound in the acute post-partum setting is accurate in making this diagnosis. To make the findings of this trial as generalisable as possible, all women whose placenta is retained following a vaginal delivery after 14 weeks gestation are eligible for inclusion. This means that women having a miscarriage, livebirth and stillbirth are all eligible for inclusion.

C9: Exclusion Criteria: Clarify how you will ascertain the absence /presence of these excluding factors. Bleeding should be an exclusion criteria because this is an indication to intervene.

R9: The decision to include a woman in the trial is the responsibility of the attending clinician. An on-site clinician makes an evaluation of blood loss and should they have any concerns about haemodynamic instability due to blood loss, women are not eligible for inclusion.

C10: Pages 10,11. All the paragraph describes the consent rather than recruitment. You refer to risk factors for RP without mentioning these risk factors.

R10: Thank you for your comments. We have inserted a comment about the scheduled timescale of recruitment and also described the clinical setting from where our subjects are recruited from. In addition, we have also highlighted risk factors for retained placenta (lines 226-228).

C11: Randomisation and allocation. All the paragraph about allocation and not randomization. What is method of randomization, do you make stratification by center.

R11: As requested, we have added the method of randomisation in the revised manuscript (lines 258-261).

C12: Follow up procedures. Page 13:

C12.1: You have to describe the intervention separate from follow up. You will include cases of RP after vaginal birth and miscarriage. You will include cases managed actively or expectantly in the third stage. You have to describe what you will do in each clinical scenario from the time of diagnosis, then the intervention and the follow up of cases. I wonder if you will examine the cases vaginally to see if the placenta is detached or not.

R12.1: As requested, we have now split the description of the intervention from the follow-up (sections 2.3.4 and 2.3.5). These sections now confirm cases managed both actively and expectantly are included in the trial. The trial design has similar study intervention and follow-up for all clinical scenarios, which have led to a diagnosis of retained placenta. As the design of the trial is a pragmatic one, the protocol does not stipulate that a vaginal examination is required to check as to whether the placenta has detached or not. Normal clinical practice should ensure that a final check is made to verify that the placenta cannot be delivered before making the decision to take the woman to theatre.

C12.2: You will do ultrasound examination or not.

R12.2: As detailed in R8, we will not do an ultrasound.

C12.3: The intervention will be NTG spray only or there will be oxytocin drip.

R12.3: As detailed, the intervention will be GTN spray only. Clinicians are able to use other treatments such as oxytocin drip according to their local protocols. However, although oxytocin reduces bleeding, there is no evidence that it is effective in treating retained placenta.

C12.4: Specify the Intervention, manufacturing company etc.

R12.4: We have specified what the intervention is: glyceryl trinitrate, peppermint oil BP 88, 1,1,1,2-tetrafluoroethane (HFA), and ethanol BP (96%) and placebo. The placebo is identical to the active product minus the glyceryl trinitrate.

In addition, we have provided details of the manufacturer (Pharmasol Limited) and labeler (Sharp Clinical Services (UK) Limited (lines 255 – 256).

C12.5: I wonder if there will be any uterine massage after delivery of the baby.

R12.5: Clinicians are allowed to follow their normal practice, which may include uterine massage following delivery of the baby.

C12.6: The time limits mentioned will be applied to cases of RP following miscarriage.

R12.6: The same time limits will be applied regardless of when the placenta is retained.

C12.7: I wonder if you will measure blood loss after delivery of the baby till delivery of the placenta or not.

R12.7: The study protocol requires the blood loss to be documented from the time the intervention is delivered until transfer to the post-natal area. Clinical staff are also required to note the total blood loss from the delivery of the baby to the delivery of the placenta.

C12.8: I wonder if starting breast feeding immediately (before delivery of the placenta) will be allowable.

R12.8: Breast-feeding is allowed before delivery of the placenta.

C12.9: You will include two types of questionnaire (one related to safety and the other to satisfaction), either mention details related to content, or alternatively add as annex.

R12.9: As requested, the questionnaires have been added as an annex.

C13: Page 14: Definitions and Details of reporting safety issues are mentioned in the manual of operations of the trial or SOPs. In the protocol, you put that you are following GCP procedures with a reference.

R13: We are not entirely sure what the referee is referring to. We feel that it is important to detail the safety reporting procedures for complete transparency for the trial.

C14: Primary outcome measures

C14.1: Page 20:[The primary safety outcome is measured blood loss between administration treatment and transfer to the postnatal ward or other clinical area.]. Mention how you will measure the blood loss in the follow up procedures.

R14.1: Blood loss will be collected as per normal clinical practice. The blood loss will be measured prior to the administration of the study drug and the total blood will also be measured prior to the woman being moved from the birthing area. The difference will be calculated (total loss minus loss measured before the study drug is given) and recorded as source data.

C14.2: Page 21:[The primary economic outcome]. Mention more details on how to calculate this outcome

R14.2: As requested, further details have been provided regarding sources of unit costs and how they are calculated (lines 576-583).

C15: Secondary outcomes:

C15.1: Manual removal of the placenta is a primary outcome

R15.1: We have defined the “need” for manual removal of placenta as the clinical primary outcome.

The secondary outcomes are divided into clinical and cost related outcomes. One of the secondary clinical outcomes is manual removal of placenta in theatre which we believe differs from the primary clinical outcome which is the “need” for manual removal of placenta. Thus the secondary clinical outcome of manual removal of placenta in theatre is actually a subset of the primary clinical outcome (that is, those who went on to actually have the manual removal of placenta – it is possible for some women that having decided there is a need for manual removal, it doesn't actually need to be done, since the placenta comes away before reaching theatre or the operation being performed).

C15.2: Use of general anesthesia is linked to manual removal of the placenta or you will use it for other condition.

R15.2: The use of general anaesthesia will be recorded regardless of its indication.

C15.3: [sustained uterine relaxation after removal placenta requiring uterotonics]. You should mention what is the routine practice after delivery of the placenta, to differentiate it from additional uterotonics.

R15.3: 10 IU of oxytocin continues to be the recommendation given in the recently updated NICE pathway “Care in third stage of labour” to assist with active management of the third stage of labour. Routinely given uterotonics will be collected, as well as use of additional uterotonics therefore allowing an evaluation of additional uterotonics.

C15.4: Fall in hemoglobin and fall in blood pressure are safety issues. As you know NTG is a nitric oxide donor with an effective smooth muscle relaxant with a potent and short-lived tocolytic effect.

R15.4: We are recording these as secondary outcomes.

C15.5: Maternal pyrexia per se is not an outcome, unless it is related to infection as a consequence of MOP. It is better to put need for treatment with antibiotics.

R15.5: We collect maternal pyrexia in the first 72 hours of delivery/ or discharge from hospital whichever comes sooner as it may be an early indication of maternal uterine infection. In addition, we also collect use of antibiotics from the period 24 hours prior to study drug administration to 6 weeks after study drug administration and we intend to address antibiotic usage within the economic evaluation.

C15.6: Subsequent surgical evacuation of retained products of conception. NTG could reduce the need for MOP but increase in the need for surgical evacuation of retained products

R15.6: We are collecting this as a secondary outcome.

C16: Data Collection and Processing.

C16.1: Measuring outcomes, the title is measuring outcomes, but the content don't reflect this.

R16.1: Thank you for this observation. We have amended the title of this section to "Collection Of Outcome Data."

C16.2: Page 20: The primary safety outcome (measured blood loss between administration treatment and transfer to the postnatal ward/other clinical area) is recorded immediately prior to transfer from theatre to the postnatal ward or other clinical area (e.g. labour ward high dependency). How you will measure the blood loss

R16.2: We request the clinical staff to measure the blood loss from the time the study medication is given until the time the woman leaves the birthing area. The normal practice is to collect the blood in a measuring vessel.

C16.3: The primary patient satisfaction outcome is collected at six weeks by a self-completed satisfaction questionnaire] put the questionnaire as annex]

R: As requested, the questionnaire has been added as an annex.

C16.4: Estimated health service costs will be compared with multiple trial outcomes in a cost-consequence analysis of GTN versus standard practice. The costs and consequences of the alternative approaches to treatment will be summarised using a balance sheet, highlighting the costs and outcomes favouring each treatment option]. Put more details on this cost-sequence analysis or put reference.

R16.4: A reference to the NICE manual for developing guidelines has been incorporated, which outlines the principles of cost-consequence analysis and the use of a balance sheet approach.

C16.5: Page 22, row 504 [Outcome data is collected on electronic case report forms and paper-based questionnaires]. Your data collecting tools should be added as annex or you mention the main information collected as text.

R16.5: Thank you. We have described what our main sources of outcome data are and the time points at which they are collected. However, we have now described the main section headings of our electronic case report form as suggested in the text.

C17: Sample size. Page 22:

C17.1: Though the magnitude of effect of NTG spray per se is not known, but the effect of NTG is reported in at least three RCTs. This could be a guide to calculate the sample size.

R17.1: As discussed above, the overall number of trials (n=3), and their combined sample size (n=175), and the perceived low quality of the evidence base meant there was little support for any specific assumption about the magnitude of the treatment effect we might reasonably expect. Instead, we considered that due to the low cost of the drug, quite a small treatment effect would be worthwhile (in terms of cost effectiveness), tempered by the difficulty of the setting (in terms of the very limited time window to intervene) and the need to ensure the women are not at risk of bleeding (taking heart rates) creates additional burdens for the staff and hence a larger treatment effect would be needed to justify this additional activity. Coupled with the additional uncertainty around the untreated event rate, we decided to assume this control rate would be 50% (which from a statistical perspective is the value with maximum binomial variability) and then assume a 20% relative reduction (an absolute reduction of 10%, from 50% to 40%) would be an appropriate magnitude of effect that would, if demonstrated, be sufficient to change clinical practice. In discussions with clinicians around the GOT-IT sites, and with the NIHR/HTA Commissioning Board, and with the flexibility of the group sequential design to declare the question resolved earlier if we had a larger treatment effect and/or less variability, this was felt to be a sensible approach given the lack of evidence to inform the choice.

C17.2: What is the software, you will use for data entry and analysis?

R17.2: We have included what software we will use in the Statistical analysis section.

C17.3: What statistical methods you will use for different comparisons.

R17.3: We have added generalised to linear models appropriate to the distribution of each outcome which describes what methods we will use. There will be a comprehensive Statistical Analysis Plan (SAP), which will govern all statistical aspects of the study, including the statistical methods for the analysis of all the outcomes, approaches to missing data, and sensitivity type analyses to investigate the robustness of any findings reported. The current draft of the SAP is at the moment being reviewed by the independent Trial Steering Committee and will be finalized before the data are locked for final analysis, and will be freely accessible on the CHaRT (study CTU) website or on request.

C17.4: What is the level of significance?

R17.4: We will use the conventional level of significance of $P < 0.05$. If the finalised SAP includes pre-specified subgroup analyses of the primary outcome, these will be considered against a stricter level of significance of $P < 0.01$, to guard against over interpretation of the data. As indicated in 17.3, the SAP is currently under review by the Trial Steering Committee and will be finalized prior to the data being locked for final analysis.

C17.5: Page 24, row 562 Specify what subgroup analyses you will explore.

R17.5: As indicated in 17.3, the SAP is currently in draft and has not been finalised. At present we are not planning to include any formal subgroup analyses of the primary outcome, however this may change depending on feedback from the Trial Steering Committee. If we decide to include subgroup analyses this decision will be made before the data is locked for final analyses. In this event, we will formally assess subgroup(s) by fitting a subgroup * randomized group interaction term and these interactions will be assessed against a stricter level of significance of $P < 0.01$ to guard against chance findings.